# Millimeter-sized smart sensors reveal that a solar refuge protects tree snail *Partula hyalina* from extirpation

Cindy S. Bick[1,5], Inhee Lee[2,4,5], Trevor Coote[3,6], Amanda E. Haponski[1,6], David Blaauw[2✉] & Diarmaid Ó Foighil [1✉]

Pacific Island land snails are highly endangered due in part to misguided biological control programs employing the alien predator *Euglandina rosea*. Its victims include the fabled Society Island partulid tree snail fauna, but a few members have avoided extirpation in the wild, including the distinctly white-shelled *Partula hyalina*. High albedo shell coloration can facilitate land snail survival in open, sunlit environments and we hypothesized that *P. hyalina* has a solar refuge from the predator. We developed a 2.2 × 4.8 × 2.4 mm smart solar sensor to test this hypothesis and found that extant *P. hyalina* populations on Tahiti are restricted to forest edge habitats, where they are routinely exposed to significantly higher solar radiation levels than those endured by the predator. Long-term survival of this species on Tahiti may require proactive conservation of its forest edge solar refugia and our study demonstrates the utility of miniaturized smart sensors in invertebrate ecology and conservation.

[1] Museum of Zoology and Department of Ecology & Evolutionary Biology, University of Michigan, Ann Arbor, MI, USA. [2] Department of Electrical Engineering & Computer Science, University of Michigan, Ann Arbor, MI, USA. [3] Partulid Global Species Management Programme, Tahiti, Polynésie Française. [4]Present address: Department of Electrical & Computer Engineering, University of Pittsburgh, Pittsburgh, PA, USA. [5]These authors contributed equally: Cindy S. Bick, Inhee Lee. [7]Deceased: Trevor Coote. ✉email: blaauw@umich.edu; diarmaid@umich.edu

Pacific Island land snails form an inordinate share of known and ongoing extinctions[1–3]. Human agency has been their primary extinction driver, mainly through the introduction of snail predators, e.g., the North American rosy wolf snail, *Euglandina rosea*[4,5]. This predator has been implicated in the extinction of at least 134 snail species[2], many of them members of the scientifically prominent[6–9] Society Island *Partula* species radiation[10–12]. Although this radiation is now largely extinct, a few representatives still persist in the wild either in montane refuges or as relictual valley populations[12–15]. One Society Islands species, *Partula hyalina*, exhibits by far the best low altitude survival with small, scattered populations enduring in 31 Tahitian valleys[13]. Its ability to survive >40 years of predation by *Euglandina rosea* has been attributed to its higher clutch size as well as to a hypothesized species-specific solar refuge[13].

Ambient sunlight, along with temperature and humidity, plays a major role in the behavioral ecology of land snails[16,17]. These organisms generally restrict their activity to low-light environmental conditions[16], thereby shielding their water-permeable skins[18] from the thermal and desiccation stresses associated with high intensity solar radiation[17]. Land snail eyes can register the direction of ambient light intensities and most species exhibit negative phototaxis, actively moving down light gradients from sunlit areas to nearby shaded microhabitats[16]. Shade is less available in more open, sunlit, environments and land snails with high albedo shell coloration (that reflect most of the light in the visible spectrum thereby reducing incident solar radiation thermal stress) appear have an enhanced ability to colonize such environments[17–22]. That facility may also apply in the case of the surviving Tahitian species, *Partula hyalina*. Early malacologist H. E. Crampton noted that in addition to occurring in dense forest (the typical partulid habitat) he observed most individuals of this distinctly white-shelled species in forest edges and stream border habitats where "larger forest trees are fewer and there is more sunlight"[6].

The solar refuge hypothesis[13] predicts that surviving populations of *P. hyalina* will be restricted to forest-edge habitats where the ambient solar irradiation conditions are, during periods of maximum sunlight, significantly higher than those tolerated by foraging *Euglandina rosea*. This hypothesis is consistent with Hawaiian field studies showing that *Euglandina rosea* exhibits higher activity levels and population densities in shaded, moist habitats[23]. If the predator is unable to engage in continuous hunting behavior in sunlit forest-edge environments tolerated by *P. hyalina*, this environmental effect may play an additive role, along with larger clutch sizes[13,14], in the latter's differential survival.

Our goal in this study was to test the predictions of the solar refuge hypothesis in the field by using light sensors to characterize the respective solar ecologies of *Partula hyalina* and of *Euglandina rosea* in Tahitian valleys. Biologists have deployed a variety of sophisticated light sensors in recent years to address diverse research questions[24,25]. However, most sensor systems use commercial off-the-shelf packaged chips, and construction of a smart solar sensor in this way would result in ~μW power consumption, requiring a 5-mm battery[26] and resulting in systems >12 × 5 × 4 mm. Therefore, a key challenge in this study was designing and constructing a sensor small enough not to interfere with the movement of foraging *E. rosea* (adult shell length 3–7 cm). To achieve this, we modified the Michigan Micro Mote (M³)[27], a smart sensing platform ~2 × 5 × 2 mm in size (including rugged encapsulation) that has been reported as the world's smallest computer[28].

## Results and discussion

Using the custom developed smart solar sensors, three field populations of *P. hyalina* and two of *E. rosea* were investigated in August 2017 (Fig. 1a). *P. hyalina* individuals were restricted to forest-edge habitats (Supplementary Fig. 1a-c), where they aestivated during daylight hours, attached to the undersides of leaves

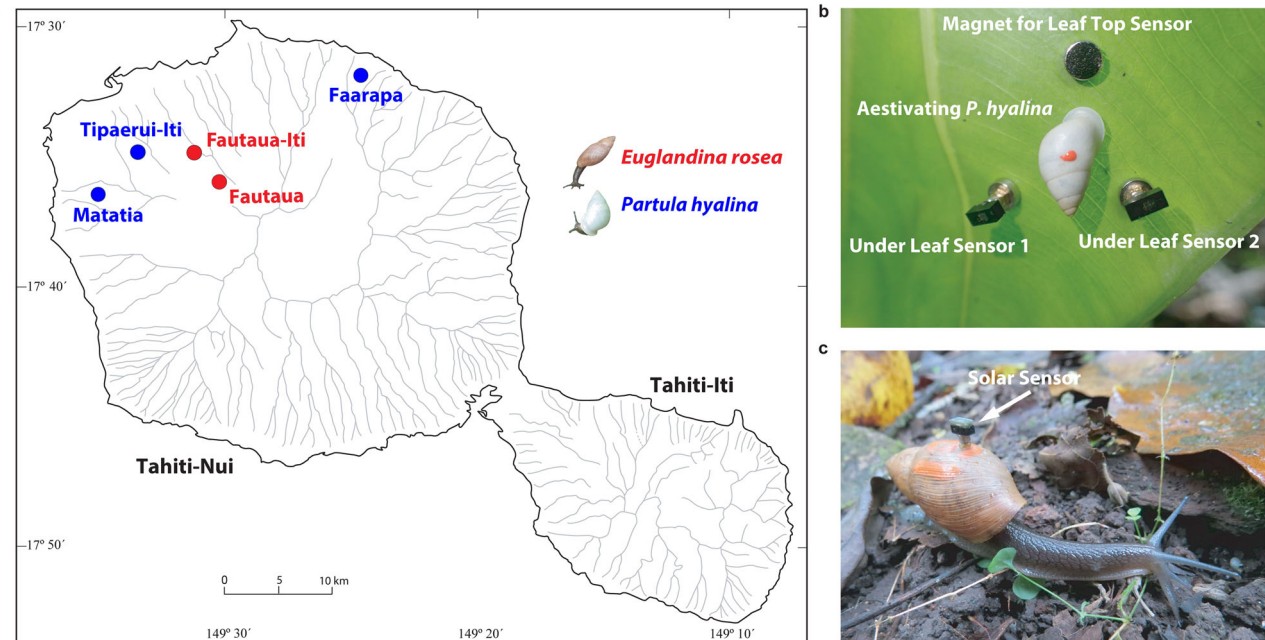

**Fig. 1 Sampling locations and smart sensor placement for the endangered Tahitian tree snail *Partula hyalina* and the invasive predator *Euglandina rosea*. a** Map of Tahiti showing the valley sampling locations for the endemic Tahitian tree snail *Partula hyalina* (green circles) and the invasive alien predator *Euglandina rosea* (red circles) study populations. **b** An aestivating specimen of *P. hyalina* attached to the underside of an *Alocasia macrorrhiza* leaf, Faarapa Valley study site. Our permit did not allow us to attach a sensor directly to the endemic species so the tree snail is flanked by two light sensors measuring its immediate (under leaf) environment and by a magnet anchoring an upper leaf light sensor measuring the leaf's ambient (leaf top) environment. **c** A foraging *E. rosea* specimen bearing an attached solar sensor, Fautaua Valley study site.

(Fig. 1b) of a variety of host plant species. Our permit did not allow the direct attachment of light sensors to this endangered species. Using magnets, we placed sensors adjacent to aestivating tree snails to record their immediate light environments (Fig. 1b) and also on the upper surfaces of the supporting leaves to record their ambient light environments (that attacking predators would have to transit).

Robust populations of *E. rosea* were present in both of its study sites (Fig. 1a) where we targeted actively crawling adult predators with ready access to both shaded and open habitats (Supplementary Fig. 1d, e). One initial concern was that the process of handling the predators might noticeably change their behavior. However, specimens equipped with attached light sensors (Fig. 1c) promptly resumed active crawling (see Supplementary Movie 1) and covered linear distances of ≤5 meters prior to data recovery in the late afternoon, with a large majority of this time spent in undergrowth, especially during sunny conditions. We inadvertently got another opportunity to test for handling effects in the case of a Fautaua-Iti specimen equipped with a light sensor on August 8th that eluded recapture until August 11. This snail's sensor yielded broadly similar light intensity profiles for days 1 and 2 (Fig. 2a), the sensor's recording time limit, implying that major handling effects were unlikely. The day 2 light profile (Fig. 2b) is of particular interest because it tracks a free-ranging predator through a full solar day. In the interval between the sun clearing the valley walls (~9:20 am) and reaching peak irradiation values (12:00–1:00 pm), the predator experienced three cycles of increasing, then decreasing, light intensities (Fig. 2b). We infer from these data that the predator sought progressively greater cover as the ambient sunlight intensified, keeping its exposure to <900 lux. It found deepest cover (down to 0 lux) during midday peak radiation levels and remained under cover for the rest of the solar day (Fig. 2b).

Figure 3 summarizes the light intensity data recorded for our Tahitian study populations of *E. rosea* (2 sites and 37 specimens over 4 days) and *P. hyalina* (3 sites and 41 specimens over 4 days). The two species experienced highly distinctive daily solar irradiation profiles (Fig. 3a-e). This was especially apparent for the leaf top sensor data. Leaves supporting aestivating *P. hyalina* experienced much higher mean ambient irradiation levels than did the predators throughout much of the solar day (10:00 am–3:30 pm; Fig. 3b). That difference was maximal around noon (12:10–12:50 pm) with ≥10-fold higher mean readings for leaf top sensors (7674–9072 lux) relative to those attached to *E. rosea* individuals (540–762 lux). The *P. hyalina* under-leaf sensors also recorded higher light intensities than the predator-attached sensors for much of the solar day (11 am–3 pm), with a maximum 7-fold mean difference (4415 versus 606 lux) recorded at 12:40 pm (Fig. 3b). A repeated measures ANOVA of the entire field dataset demonstrated that the *P. hyalina* leaf top, *P. hyalina* under leaf, and foraging *E. rosea* sensor readings all differed significantly in their solar radiation levels (Table 1).

Previous research into the abiotic ecological factors limiting *E. rosea*'s effectiveness as a predator have focused on temperature and humidity[23,29,30] but our results demonstrate that environmental light intensity may be an additional factor. Although one of the sensor-equipped predators briefly endured 13,449 lux of sunlight while crossing an open trail (Supplementary Fig. 2), our data indicate that foraging predators rarely exposed themselves to >3000 lux (Fig. 3e; Supplementary Fig. 3). In contrast, forest-edge host plants harboring aestivating *P. hyalina* were exposed to mean solar irradiation intensities >3000 lux throughout much of the solar day (10:10 am–2:00 pm), reaching a peak mean of ~9000 lux at midday (Fig. 3b), with individual host plant leaf top sensors recording up to 71,165 lux (Fig. 3c).

Any solar refuge effect experienced by the surviving *P. hyalina* populations is necessarily intermittent, being limited to daylight hours and being attenuated by overcast conditions (Supplementary Figs. 4, 5). Our predator field data (Fig. 3; Supplementary Fig. 3) imply that light intensities >3000 lux may be required to effectively deter foraging *E. rosea*. The protective effect at the Matatia Valley site on an overcast day was marginal at best and, at the Tipaerui-Iti Valley site, it fluctuated markedly over the

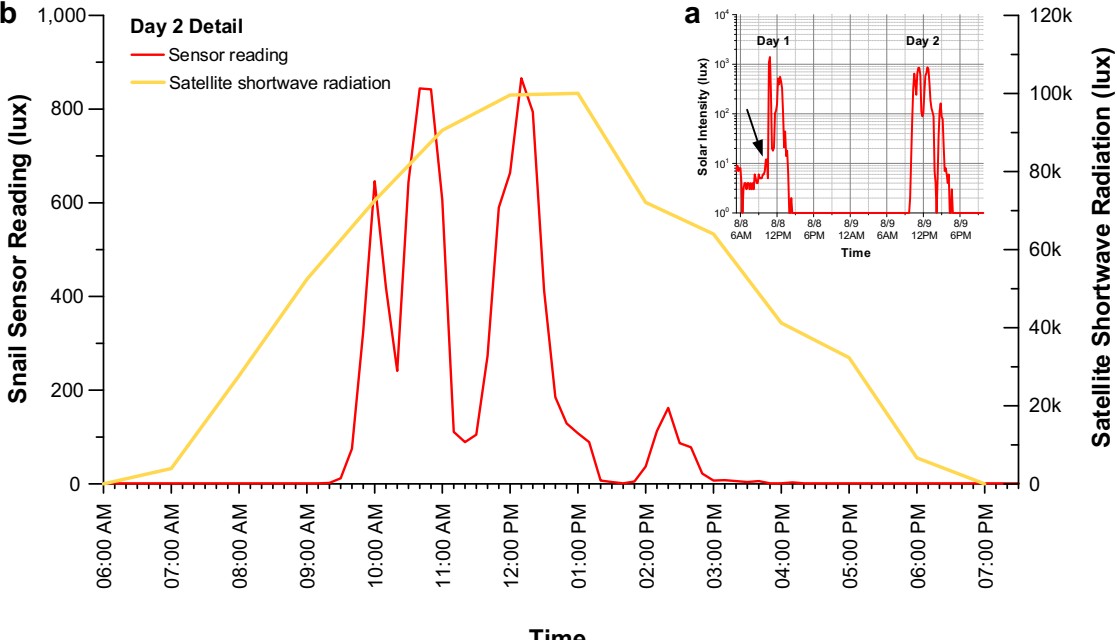

**Fig. 2 Solar ecology readings for a single Fautaua-Iti specimen of *Euglandina rosea* over two days. a** The predator's light intensity profile is shown for August 8th (Day 1) 2017 and, after it eluded recapture, for August 9th (Day 2), 2017. The black arrow indicates when *E. rosea* was released after the sensor attachment. **b** A detailed view of the predator's August 9 light intensity profile superimposed with that day's record of Tahitian solar irradiation levels, measured by satellite and obtained from meteoblue[49].

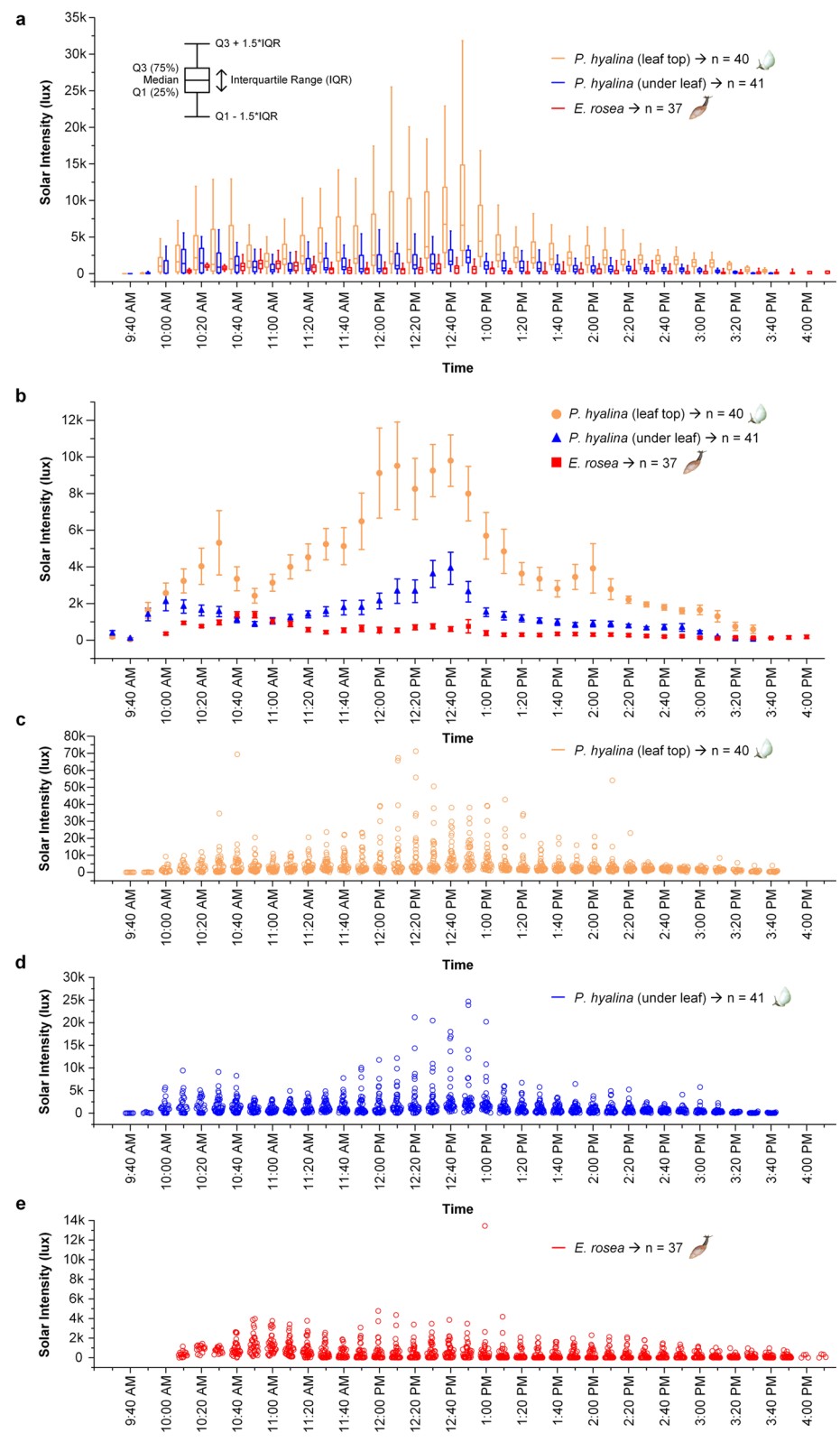

**Fig. 3 Tahitian field solar irradiation intensities recorded for the endemic tree snail *Partula hyalina* and the alien predator *Euglandina rosea*.** Light intensity field data were integrated over 10-min daily time intervals for 37 biologically independent *E. rosea* specimens (red lines) among two localities over 4 days, and for 41 (leaf top, orange lines) and 40 (under leaf, blue lines) biologically independent aestivating specimens of *P. hyalina* among three localities over four days. **a** Boxplot of the recorded light intensities showing median, first quartile (Q1), third quartile (Q3), interquartile range (IQR), 1.5 times the interquartile range above the third quartile (Q3 + 1.5*IQR), and 1.5 times the interquartile range below the first quartile (Q1 − 1.5*IQR) values for each of the three data categories. **b** mean light intensities and standard errors recorded for each of the three data categories. **c** All individual leaf top light intensity values (orange circles) recorded for 41 aestivating specimens of *P. hyalina*. **d** All individual under-leaf light intensity values (blue circles) recorded for 40 aestivating specimens of *P. hyalina*. **e** All individual light intensity values (red circles) recorded for 37 specimens of *E. rosea*.

**Table 1 Partula hyalina leaf top, *P. hyalina* under leaf, and *Euglandina rosea* sensor readings all differed significantly in their solar radiation levels.**

| Factor | Chi-square | DF | *p*-value |
|---|---|---|---|
| Group | 79.58 | 2 | <0.0001 |
| Time | 25.23 | 1 | <0.0001 |
| Group × Time | 38.29 | 2 | <0.0001 |

| Contrast | Estimate ± se | DF | t-ratio | *p-value* |
|---|---|---|---|---|
| *P. hyalina* leaf top vs. under leaf | −1.12 ± 0.29 | 115 | −3.82 | 0.0006 |
| *P. hyalina* under leaf vs. *E. rosea* | 1.48 ± 0.30 | 115 | 4.94 | <0.0001 |
| *P. hyalina* leaf top vs. *E. rosea* | 2.60 ± 0.30 | 115 | 8.65 | <0.0001 |

The top four rows present the repeated measures ANOVA values for each factor showing their respective Chi-square test statistic, degrees of freedom (DF), and *p*-value. The bottom four rows display the post-hoc Tukey's Test results for *P. hyalina* leaf top, *P. hyalina* under leaf, and *E. rosea* groups including the Tukey's test estimate ± standard error (se), degrees of freedom (DF), t-ratio, and corresponding *p*-value.

2 days of observation (Supplementary Fig. 4). Adult *E. rosea* mean crawling rates of ~0.3 meters per hour[29] imply that an overcast day together with its adjoining nights (~36 h) could allow predators to penetrate >10 m into a solar refuge. However, *E. rosea* movement in the field is predominantly nonlinear with observed mean resultant movement of <2.5 m per week[23]. This latter estimate is consistent with our observations that the *P. hyalina* solar refuge protective effect may operate on spatial scales as small as 1–3 m, e.g., the largest known surviving Tahitian population (hundreds of adults) occurs on a linear stand of wild red ginger *Etlingera cevuga* in Tipaerui-Iti Valley but is absent from the immediately adjacent forest (Supplementary Fig. 1a).

Our results corroborated the main predictions made by the solar refuge hypothesis[13]. Crampton's[6] interior forest *P. hyalina* populations are apparently extirpated: they remain undetected despite 13 years of Tahitian valley surveys. In addition, surviving *P. hyalina* were restricted to forest-edge habitats where they were exposed to significantly higher ambient solar irradiation levels than were foraging predators in similar habitats (Fig. 3). Nevertheless, there are a number of important caveats associated with our results that must be kept in mind. The most obvious is that our data came from separate populations of *P. hyalina* and of the predator: we were unable to find co-occurring populations (see Methods) and our permit did not allow us to experimentally translocate either species. Additionally, our permit allowed us to physically attach sensors to the mobile predator (Fig. 1c) but not to the aestivating *P. hyalina* (Fig. 1b) and it is not clear how this difference may have affected data quality. Nevertheless, the aestivating (Fig. 1b) tree snails' lack of movement during the recording time window likely allowed us to accurately assay its immediate light environment and released predators promptly resumed the active locomotion that is an integral part of their normal hunting behavior[31] although we had no information on their prior individual feeding histories. Another shortcoming was the lack of data on the study snails' ambient temperature and humidity, additional environmental factors that along with solar radiation levels influence land snail activity levels[17,18]. Ideally, we should repeat this study on co-occurring populations of these two snails using sensors that would provide multi-day recordings of these three key environmental factors.

Despite these caveats, our new data do provide additional insights into this species' unexpected survival: predation models predicted partulid extirpation within 3 years of initial predator contact[30] yet many *P. hyalina* Tahitian valley populations have survived >40 years of exposure to the predator. Our results, although correlational rather than causal, are consistent with the hypothesis that small-scale forest-edge solar refuges do exist for this tree snail species in Tahitian valleys and that, together with this species's higher reproductive output[13,14], they may have contributed to its differential survival. There are apparent parallels here with the differential survival of a subset of Guam's endemic avifauna following introduction of the brown tree snake *Boiga irregularis*: the ability to nest in snake-inaccessible locations, and/or to produce large clutch sizes, have enabled some endemic birds to endure[32,33].

Our results, together with earlier Tahitian valley survey data[34], highlight the importance of stream edge *Etlingera cevuga* (wild red ginger) stands in the continued survival of *Partula hyalina* on the island. The Tipaerui-Iti Valley site, in particular, deserves follow up detailed ecological study and characterization in an effort to identify what factor(s) have enabled it to support an exceptional population of *P. hyalina*. One potential modulating factor may be the presence/absence of leaf litter, a known preferred microhabitat for *Euglandina rosea*[23]. It is possible that presence of a well-developed leaf litter within solar refuges could undermine their protective effect by allowing predators to shelter in situ during daylight hours. Our results also indicate that long-term proactive conservation planning for *Partula hyalina* should involve mapping, protection and maintenance of wild red ginger stands in as many Tahitian valleys as possible. Active maintenance of solar refuges could involve removal of leaf litter within the stands and selective removal of encroaching tree canopies. Stands currently lacking *Partula hyalina* populations could be repopulated with transplanted individuals, ideally from the same valley. Research is also required on the solar ecology of the more recently introduced Tahitian snail predator, the New Guinea flatworm *Platydemus manokwari*[12,35], to determine if forest-edge remnant snail populations are also protected from it. With proactive maintenance of solar refuge habitats, it may be possible to ensure the indefinite survival of *P. hyalina* on Tahiti.

## Methods

**Smart solar sensor design.** To prevent interference with the movements of the highly mobile *E. rosea* predators, we developed a custom smart solar sensor using the Michigan Micro Mote (M³) platform[27,36]. The M³ platform consists of a family of chips that can be integrated together through die-stacking in various ways, allowing its functionality to be customized. M³ achieves this degree of miniaturization by directly stacking bare-die chips, thus avoiding individual chip packaging, and custom-designed low-power circuits, reducing consumption to only 228 nW. The resulting systems can be powered for >1 week by a chip-scale battery[36] measuring only 1.7 × 3.6 × 0.25 mm. For the solar sensor, we selected chips from this set with the following functionalities and stacked them as shown in Fig. 4: (1) two custom-designed thin-film lithium-chemistry batteries[37], each with 8-μAh capacity and 4.2-V battery voltage, connected in parallel; (2) a power management chip to generate and regulate the three supply voltages used by the M³ chips from the battery supply voltage; (3) a microprocessor chip containing an ARM Cortex-M0[38] processor that executes the program controlling the sensor and 8 kB of SRAM for storing program and sensor data; (4) a short-range (5 cm) radio chip with on-chip antenna for retrieving data from the sensor; (5) a decoupling capacitor chip for stabilizing supply voltages; (6) a harvester chip that up-converts the voltage from the photovoltaic (PV) cells to the battery voltage and regulates battery charging; (7) a temperature sensor chip; (8) an inactive spacer chip that provides physical separation between the PV cell, which is exposed to light, and the remainder of the chips below it, which must be blocked from light; and (9) a PV chip for harvesting solar energy, containing also a small PV cell for receiving optical communication.

The battery chips measured 1.7 × 3.6 × 0.25 mm while the remaining chips were 1.05 mm wide, 150 μm thick and varied in length from 1.33 to 2.08 mm. The chips were stacked in staircase fashion (Fig. 4a) using die-attach film and connected electrically using wire bonding with gold 18-μm diameter wire. The radio die extended beyond the other chips at the back to expose the antenna. The chips communicated using a common bus protocol, called M-bus[36]. The final chip stack was encased in epoxy (Fig. 4b). The top portion of the sensor was encased with clear epoxy to allow light penetration, thereby enabling energy harvesting and optical communication. The bottom portion was encapsulated with black epoxy to protect the sensitive electronics from light. Finally, the entire sensor was coated

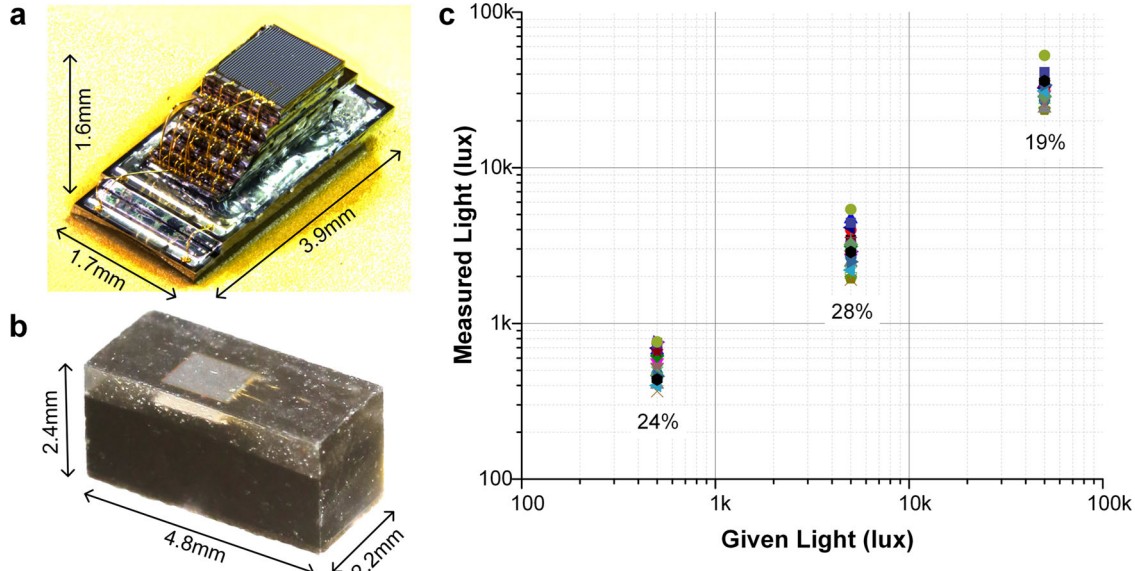

**Fig. 4 Structure and testing of custom-designed smart solar sensors. a** Smart sensor before encapsulation showing the interconnected stack of chips. **b** Smart sensor after encapsulation with black and clear epoxy. **c** Sensor readings of eight randomly selected smart sensors, each indicated by a distinct symbol, at three light intensities across the temperature and battery voltage ranges observed during the sensor deployment with the σ/μ annotated.

with 4 μm of parylene. The sensor was tested to withstand immersion in brine at pressures up to 600 atm for 1 h and in saline solution for 2 weeks.

The principal approach to reduce the M³ sensor's power consumption is to duty-cycle its operation, meaning the processor executes code briefly (ms range) every 10–60 min and is in "sleep mode" for the remainder of the time. Sleep power is highly optimized to ~100 of nW using a number of recently developed circuit techniques[39–41]. In active mode, the processor is operating and obtains and stores sensor data. The processor clock frequency was set to 80 kHz, and at 0.6 V supply, the power consumption was 1.0 μW. In sleep mode, the processor and logic are power-gated[42], and only the SRAM, timer, optical receiver, and power management remain on, reducing the power consumption to only 160 nW. The 10-min sleep mode period length was selected to amortize the high power in active mode and minimize overall power consumption while retaining a sufficiently small sensor acquisition interval for the proposed study. The average power consumption of the entire sensor including all peripherals was 228 nW, and in tests, it was able to operate on a battery charge alone for 1 week. With PV-based harvesting, the sensor becomes energy autonomous at light levels >850 lux. For this study, the sensors were retrieved and recharged using a light station after each deployment.

Although duty-cycling lowers the average current draw from the battery, it limits measurements to times when the sensor is awake. This raised a particular difficulty for measuring the solar ecology of snails where continuous light monitoring is essential, preventing the use of duty-cycling. Typical light-sensing circuits monitor the current from a photodiode and consume ~μW power[43], a load that would deplete the batteries in only a few hours. Hence the light intensity had to be monitored during sleep mode. To achieve this without substantially increasing the sleep mode current draw, we observed that the harvester circuit inherently integrates and quantizes the harvested energy from the photovoltaic (PV) cell in a manner proportion to the ambient light level and can be modified to function as a light sensor readout circuit.

To up-convert the output voltage of the PV cell (250–450 mV) to that of the battery (3.9–4.2 V), the harvester performs a series of voltage doublings[44] using the circuit shown in Supplementary Fig. 6. Each voltage doubling circuit consists of two chains of inverters, configured as a ring oscillator. The two oscillators are coupled through on-chip MIM capacitors and are connected to the supplies $V_{in}$ and $V_{double}$, as shown. During one oscillation cycle, each capacitor experiences two different configurations. When the input to its driving inverters is high, a capacitor is placed between $V_{in}$ and ground (GND), i.e., in parallel with the PV cell, which charges it with a finite amount of charge. When its driving inverter inputs are switched low, the capacitor is placed between $V_{in}$ and $V_{double}$, and it delivers the received charge to $V_{double}$, thereby up-converting the voltage from the PV cell. The amount of charge that is transferred per cycle is kept constant by the frequency regulation circuit. If the PV cell is exposed to intense light and produces a high current, the regulation circuit increases the frequency by reducing the delay of the voltage-controlled delay element to maintain a constant charge transfer per cycle. Conversely, if the light level drops, the regulation circuit slows the oscillation frequency.

As a result, the frequency of oscillation is proportional to the PV current to the first order. And, because the current of the PV cell is proportional to the light intensity, the oscillation frequency is a measure of the instantaneous ambient light level. To obtain the light dose over a sleep mode time period, we added a low-

power counter (shown in Supplementary Fig. 6), which records the number of oscillations during this period, thereby integrating its total light dose. Each active-mode period, the microprocessor reads the counter value, resulting in a light sensor code, and resets the counter. The counter operates at a low supply voltage of 0.6 V, which reduces its power consumption by ~9× compared to a standard supply of 1.8 V. This allowed us to implement a 24 bit counter with negligible power consumption (5 nW or 2.2% of total average power). The resulting sensors continuously monitor the light level and record a light-dose code for every 10 min interval. The addition of the counter constitutes a relatively small change in the harvester circuit and allows light monitoring without additional chips or an increase in battery capacity or sensor size.

**Sensor testing and calibration**. Because the harvester oscillation frequency is dependent on temperature and battery voltage, these parameters are stored by the processor in SRAM along with the light sensor code. After data retrieval, the code is then converted to light intensity using a model that accounts for the temperature and battery voltage dependency. To construct this calibration model, four sensor nodes were measured at six light levels (0.5, 1, 5, 10, 50, and 100 klux) and four temperatures (25, 35, 45, and 55 °C), and four battery voltages (3.9, 4.0, 4.1, and 4.2 V); a total of 96 measurements were made for each sensor. After averaging the light sensor codes across the four sensors, a multidimensional, piecewise linear model was used to establish the relationship between the recorded digital code and the light intensity at a particular temperature and battery voltage (Supplementary Fig. 7). To calibrate the model for each fabricated sensor, we measured the light sensor code, temperature sensor code and battery voltage sensor code in controlled conditions (temperature: 25, 45, and 55 °C; light: 5 klux; battery voltage: 4.1 V) for each sensor. We then applied three-point calibration of the temperature sensor and one-point calibration of both the battery voltage and light sensors. The calibration conditions were selected based on the expected temperature and battery operating range in the field and on what the discriminating light intensity was expected to be. This was balanced with the time required to measure the 55 deployed systems in a controlled environment.

To verify the accuracy of the light readings, eight randomly selected sensor systems were tested at three light levels (0.5, 5, and 50 klux) and three temperatures (25, 30, and 35 °C), a total of nine conditions each. These testing conditions were selected to match the conditions that sensors experienced during the field testing and are representative of the error in light readings for the collected data. Figure 4c shows the resulting measurements after calibration was applied. The x-axis is the reported light level, and the y-axis is the actual light level the sensor was exposed to. The worst-case variation in reported light measurement was sigma/mean = 28%, at 5 klux, showing acceptable stability.

Nonlinearity was worse with a sensor light reading to actual controlled light intensity ratio ranging from −37 to +14%. However, because this is a comparative study of prey and predator species, and the same individual sensors were reused for both the prey and the predators, nonlinearity was judged to be less important than sensor-to-sensor variation and variation resulting from temperature change.

We manufactured 201 smart solar sensor systems, most of which were used for bench top testing and green house testing at the University of Michigan using locally caught specimens of *Cepaea nemoralis* land snails (Supplementary Fig. 8).

A total of 55 tested units were taken to Tahiti and were reused in multiple deployments while there. Our small batch production cost for these sensors was ~ $500 US per unit (including wafer fabrication, wafer dicing, system assembly, encapsulation, and yield loss); however, for large volume (>200 units) production, this was reduced to ~$150/unit.

**Field methods**. Two field populations of *E. rosea* and three of *Partula hyalina* located in five northern valleys of Tahiti-Nui, the main Tahitian peninsula, were investigated in August 2017 (Fig. 1a). These locations were selected by T. Coote, who had conducted extensive field surveys on Tahiti since 2004, as being the most accessible populations of both species then available.

Although *E. rosea* remains widely distributed throughout Tahiti, it has become less numerous in many valleys in recent years, possibly because of the introduction of another snail predator, the New Guinea flatworm *Platydemus manokwari*[12,35]. Dead *E. rosea* shells were much more common than live specimens at our three *Partula hyalina* study locations, so we focused instead on the robust predator populations present in the nearby main Fautaua Valley and in its side-valley Fautaua-Iti. In both locations, we picked sites where foraging *E. rosea* had ready access to both shaded and open habitats. The Fautaua-Iti Valley location consisted of an open sunlit trail through the rainforest (Supplementary Fig. 1d), and the solar ecologies of nine predators were monitored here on two days: 5 on August 8 and 4 on August 11. The Fautaua Valley location consisted of a forest-edge adjoining an open grassy area (Supplementary Fig. 1e), and 29 predators were monitored here over two days: 12 on August 12 and 16 on August 14.

All three of our *Partula hyalina* study sites (Fig. 1a) consisted of discrete patches of vegetation between the edge of the forest and the primary stream, or *captage*, within each valley. The Tahitian valley of Tipaerui encompasses a small side valley, Tipaerui-Iti, which contained the most robust known surviving population of *P. hyalina* on Tahiti, consisting of hundreds of individuals. They were restricted to a linear stand of *Etlingera cevuga* extending for 60–70 m (Supplementary Fig. 1a). The solar ecologies of 28 aestivating Tipaerui-Iti *Partula hyalina* individuals were recorded over two days: 12 on August 10 and 16 on August 15. *Partula hyalina* population sizes were much smaller in the other two valleys, Faarapa, and Matatia (Fig. 1a), requiring us to monitor all of the individuals we encountered. The Faarapa Valley site consisted of a mixed stand of *Barringtonia asiatica*, *Alocasia macrorrhiza*, and *Pisona umbellifera* (Supplementary Fig. 1b). We detected six individuals at this site, and their solar environments were monitored on August 5. Our remaining *Partula hyalina* study site was in Matatia Valley (Fig. 1a), where a small, low-density population occurred in scrubby habitat attached to the foliage of *Z. officinale*, *Pisona umbellifera*, and *Inocarpus fagifer* (Supplementary Fig. 1c). A total of seven individuals were detected and assayed on August 7.

Each working day, we entered the study valley in the early morning between 8 and 9 a.m., prior to the appearance of the sun above the valley walls; and searched systematically for our respective target species. *Euglandina rosea* individuals were found foraging actively, either on the ground or climbing on vegetation, and they typically maintained this searching activity throughout the day. In contrast, *Partula hyalina* individuals were aestivating attached to the underside of leaves, and specimens typically remained in situ on the same leaf during the observation period.

To track the solar ecology of each predator, a smart solar sensor was reversibly attached to the dorsal surface of each *E. rosea* shell using a nut and screw method. The nut (McMaster-Carr, Brass Hex Nut, narrow, 0–80 thread size) was glued (Loctite, Super Glue) directly on the predator's shell, and after allowing 10 min for bonding, a sensor, preglued to a compatible screw (McMaster-Carr, 18–8 Stainless Steel Socket Head Screw 0–80 thread size, 1/16" long), was attached mechanically. Each predator was numerically labeled using nail polish and released at the exact spot it had been discovered. For the rest of the study period, each predator was visually tracked as it continued its foraging until mid-afternoon, when the sun descended below the valley walls, and the snails and sensors were recovered. These invasive predators were then euthanized.

Aestivating *Partula hyalina* attach to the underside of leaves. Because our permit did not allow the direct attachment of light sensors to this endangered species, we deployed under-leaf sensors next to the aestivating snails using a nut/ screw/magnets combination. This involved gluing, in advance, the screw to the sensor base and the nut to a round magnet (Radial Magnet Inc., Magnet Neodymium Iron Boron (NdFeB) N35, 4.78 mm diameter, 1.60 mm thickness). In the field, these components were assembled and held in place using another magnet positioned on the upper leaf surface. In addition to recording the under-leaf light intensities experienced by the aestivating *Partula hyalina* specimens, we also recorded the ambient light intensity by attaching a sensor to the upper surface of the leaves harboring the aestivating specimens.

Each working day, the data recording function of the smart sensors was activated before going into the field and was terminated after returning from the field, and the data were then retrieved via the sensors' wireless communication link. For each sensor, the recording start time, meaningful time of the measurement start time, meaningful measurement end time, and sensor recording end time were recorded to properly calibrate the time of the recorded samples. The received raw data in digital format were then translated to time and light intensity information using a MATLAB program and the calibration data specific for that sensor.

**Statistics and reproducibility**. Recordings from each of the three categories (*Partula hyalina* leaf top, *P. hyalina* under leaf, and *Euglandina rosea*) over the 8 days of field recording were aggregated into their respective 10-min time intervals from 9:30 to 16:00 h. This recording time window avoided the early morning handling period when sensors were attached to the predator, spanned the midday period of peak solar irradiation (Figs. 2, 3), and enabled us to recover the visually tracked predators before losing them in the gathering darkness of the late afternoon valley forests. We collected light intensity measurements for 40 leaf top sensors, 41 under leaf *P. hyalina*, and 37 foraging *E. rosea* snails over the 9:30–16:00 h time period. Most aestivating *P. hyalina* ($N = 26/41$) had two under-leaf sensors bracketing the snails to record their immediate light environment (Fig. 1b) and for these individuals we used the mean light intensity of the two sensors to compare to the other two categories.

We compared the three categories (leaf top, *P. hyalina* under leaf, and *E. rosea*) for the 40 timepoints over the 9:30–16:00 h time period using a repeated measures analysis of variance (ANOVA) in the *nlme*[45] and *car*[46] packages in R v.3.5.0[47]. We first tested the light intensity measurements for conformance to a normal distribution using the R code *shapiro.test*, with the result being a highly skewed distribution. We thus LOG transformed the measurement data. We specified the following linear mixed model for the 9:30–16:00 time interval using the *nlme* package in R:

$$lme(\text{LOG fullmean} \sim \text{group} + \text{time} + \text{group*time}$$
$$\text{random} = \sim 1|\text{individual},$$
$$\text{correlation} = corAR1(\text{from} = \sim \text{time}|\text{individual}$$
$$\text{method} = "REML", \text{na. action} = \text{na.exclude})$$

Where LOGfullmean = the LOG transformed light intensity readings, group = leaf top, *P. hyalina* under leaf, or *E. rosea*, time = the 40 10-min time intervals from the 9:30–16:00 time period. We considered each individual as a random block and included the correlation between time and individual. The repeated measures ANOVA utilized the restricted loglikelihood (REML) method and excluded any missing timepoint measurements (na.action = na.exclude) from the 9:30–16:00 h time period. After running the linear mixed model in R, we then used the *Anova* command from the R package *car* followed by a post-hoc Tukey's test to determine which categories significantly differed in their light ecologies.

**Reporting summary**. Further information on research design is available in the Nature Research Reporting Summary linked to this article.

## Data availability
The datasets generated and/or analyzed during the current study are stored in, and available through, the online figshare depository[48]. Any remaining information can be obtained from the corresponding author upon reasonable request.

## Code availability
Custom code and algorithms used for analyses in this study are stored in, and are available through, the figshare depository[48].

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

## Acknowledgements

We thank the Délégation à la Recherche (de la Polynésie Française) for permitting our field work on Tahiti, Jamie Phillips and Eun Seong Moon for assistance in sensor development, Mary Leys for providing *Cepaea nemoralis* specimens, Ashley Franklin for statistical advice, and three reviewers, Liew Thor Seng, Menno Schilthuizen and Paulo de Souza, for their constructive comments that significantly improved the manuscript. We gratefully acknowledge our late colleague Trevor Coote's extraordinary dedication, over many years, to Society Island partulid conservation (Supplementary Fig. 9). Financial support was provided by the University of Michigan's Office for Research Mcubed program, by a Department of Ecology & Evolutionary Biology Block Grant to C.S.B., and by NSF Award CNS-1111541 and Arm Ltd. funding to D.B.

## Author contributions

The study represents part of C.S.B.'s doctoral thesis. The experimental question was conceived by C.S.B. and D.ÓF. and the redesign of the M³ sensor was conceived and directed by I.L. and D.B. C.S.B., D.ÓF., I.L., and D.B. were all involved in testing the sensors at the University of Michigan and in designing the field experiments on Tahiti. C.S.B., T.C., and I.L. performed the field experiments and C.S.B. and I.L. led the data gathering and data analyses. A.H. performed the ANOVA analyses. All six authors participated in the writing.

## Competing interests

The authors declare no competing interests.
