## [Peer Review File · Communications Biology]

Reviewers' comments:

Reviewer #1 (Remarks to the Author):

By Liew Thor Seng

Date: 12 August 2019

This paper by Bick et al. proposed and tested a new hypothesis – solar refuge for snails (*Partula hyaline*) under predation pressure (*Euglandina rosea*). Authors developed a mm-sized light sensor that make field experiment to test the hypothesis possible. The results revealed association of the light exposure with the behaviour of the snails and its predator. The technical breakthrough of the light sensor would inspire ecologists to think and conduct similar kind of behaviour study in relation to light intensity. The solar refuge hypothesis that proposed in this study is rather novel for land snail ecology study. Hence, this article will be of interest to other malacologists and ecologists in general. The work is convincing with rigorous experimental design, data and analysis.

However, biological significant for this research need to be strengthen more. Large part of text in the article was dominated by rational and development of light sensor. For example, the discussion for the biological significant of the finding was rather short (Line 164-174) as compare with the background of micro sensor (Line 64-104). The technical development of the sensor is important for this study but the biology insight that provided by this research would better fit to the scope of Communications Biology. I suggest author refocussing this article to the biology findings.

In addition, please find other comments below. I hope these are useful for author to improve their manuscript.

Point 1. Probably it is also good to mention the cost of this sensor.

- The develop of this micro sensor would excite many ecologists to start to rethink their experimental designs and to develop interesting ecology questions related to solar influence on organisms. Thus, the feasibility (i.e. costs) would be an important aspect for ecologists to consider.

Point 2. "solar-refuge-dataset-meansensor.csv" was not available.

- Different datasets have been provided in an Excel file. However, the csv file mentioned above (in the R script) was not readily available. It is better to provide the csv file than letting reader to locate the data among spreadsheets in Excel file.

Point 3. Headings and subheadings were missing all together between Line 47 and Line 174. the manuscript.

- It is hard to for readers to navigate around the article, there are two different topics (ecology of snails and light sensor) present together here.

Point 4. Line 164-174 - "...are physiologically limited by moisture availability and are mostly found in shaded areas."

- Probably a bit more discussion of literature review required to establish the effect of light exposure to the moisture and temperature for the location/habitat.
- This study proves the association between the light intensity and behaviour of the snails but there was no causation can be proved with the existing data.
- Light intensity might not be the direct factor. Temperature, humidity and light intensity are not mutually exclusive factors (See also line 140-142). Further discussion on this would be important.

Point 5. Line 170 – 171 - "Long-term conservation planning for *Partula hyalina* should involve mapping, protection and maintenance of suitable forest edge habitats in as many Tahitian valleys as possible."

- Probably this need to be elaborated more regarding to how the finding (1) can improve current conservation strategies.

Reviewer #2 (Remarks to the Author):

This paper describes a novel way of testing the hypothesis that an invasive predatory snail fails to drive to extinction a native, threatened non-predatory snail because both have different tolerance ranges for exposure to sunlight. The authors have tested this hypothesis by applying a novel, miniature sensor, small enough to be carried by these snails on their shells.

The results of the study certainly suggest that, indeed, *Partula* escapes predation and extinction because it is able to withstand greater insolation than its predator.

The main concern I have with the paper is that the snails' activity was measured in two different ways: in *Euglandina*, the sensors were placed on the snails themselves, whereas in *Partula*, for permit restriction reasons, they were placed near resting snails. Also, if I understand correctly, the measurements were not taken in a habitat where the two species occur together, but in separate locations. This weakens the conclusions somewhat.

In addition, I have a number of minor, mostly technical comments:

- in various places in the paper, sizes are given as $A \times B \times C$ mm³. I would prefer to use either dimensions ($A \times B \times C$ mm) or volume (X mm³). A similar remark applies to surface areas ($A \times B$ mm²).
- I am surprised the paper does not refer to earlier experiments in *Cepaea nemoralis* by Jones (1982), who used a light-sensitive paint to determine the amount of time individuals spent exposed to sunlight (Reference: Jones, J.S., 1982. Genetic differences in individual behaviour associated with shell polymorphism in the snail *Cepaea nemoralis*. *Nature*, volume 298, pages749–750.)
- Line 76: replace "and" by "that"
- Line 77: "principal" instead of "principle"
- Lines 70-100 are filled with a lot of technical detail about the sensors that were developed. While I appreciate that the new technology requires explanation, perhaps this is a little overdone for the main body of what otherwise is a biology, not an electronics paper.
- I think the main text should also explain (as is done in the Methods section) why sensors were not attached to *Partula* snails, but only to *Euglandina* snails.

Menno Schilthuizen, August 26th, 2019
(I waive anonymity.)

Reviewer #3 (Remarks to the Author):

Dear authors,
thank you for your submission to *Communications Biology*. The article demonstrates the use of a miniaturized sensor to determine the amount of environmental light intensity, which fits well into the hypothesis of exposure to light be a strategy adopted by preys to seek refuge from predators. Considering the size of the animals, the sensor is a perfect solution as it is small (and I believe light).

The article is well-structured, methods well-described, references are relevant and figures overall good.

I am supportive of this manuscript to be published, provided two small points could be clarified by the authors:

1. Is environmental light intensity the key for survival? Do snails forage at night?

Rationale: I would appreciate if the authors could elaborate more on this. Light incidence is clearly distinctive, but it could be a response to other parameters already reported in the literature (e.g., temperature and humidity). If so, it could make your initial hypothesis ill-designed when solar refuge is not a unique factor, but the environmental conditions including temperature, humidity, wind speed, etc. Therefore, unless more robust arguments are presented in response to this first point, the hypothesis must be reformulated, title and claims changed accordingly.

2. Consider limitations of the proposed methods. For example:

Weight of the sensor on small snails, smell of pigments or glue in the predator behaviour, is only a few days enough? Age of the animals used in the experiments? When the predators last ate and how frequently they eat?

Overall the manuscript is well-written. I would only suggest removing adjectives such as "potent" (line 142), consistent use of commas to express numbers above 1,000 (e.g., check lines 147 and 148) and the constant use of "due" and "due to" instead of less colloquial terms such as "resulting from" and "because of".

I hope the authors would find my considerations useful and will address the points I raised and recommendations I provided.

Best regards,

Paulo de Souza

Referee expertise:

Referee #1: land snail ecology and conservation

Referee #2: snail ecology

Referee #3: micro-sensors, environmental monitoring

Reviewers' comments:

Reviewer #1 (Remarks to the Author):

By Liew Thor Seng

Date:12 August 2019

This paper by Bick et al. proposed and tested a new hypothesis – solar refuge for snails (*Partula hyaline*) under predation pressure (*Euglandina rosea*). Authors developed a mm-sized light sensor that make field experiment to test the hypothesis possible. The results revealed association of the light exposure with the behaviour of the snails and its predator. The technical breakthrough of the light sensor would inspire ecologists to think and conduct similar kind of behaviour study in relation to light intensity. The solar refuge hypothesis that proposed in this study is rather novel for land snail ecology study. Hence, this article will be of interest to other malacologists and ecologists in general. The work is convincing with rigorous experimental design, data and analysis.

Thanks for the kind words.

However, biological significant for this research need to be strengthen more. Large part of text in the article was dominated by rational and development of light sensor. For example, the discussion for the biological significant of the finding was rather short (Line 164-174) as compare with the background of micro sensor (Line 64-104).The technical development of the sensor is important for this study but the biology insight that provided by this research would better fit to the scope of Communications Biology. I suggest author refocussing this article to the biology findings.

We have taken this advice and moved the sensor development details to the terminal Methods section thereby freeing up the Introduction and Results/Discussion sections to focus on the biology. We have beefed up the biological component by adding

significant new text to both Introduction and Results/Discussion sections (highlighted in the revised manuscript) in addition to 9 new biological references.

In addition, please find other comments below. I hope these are useful for author to improve their manuscript.

Point 1. Probably it is also good to mention the cost of this sensor.

- The develop of this micro sensor would excite many ecologists to start to rethink their experimental designs and to develop interesting ecology questions related to solar influence on organisms. Thus, the feasibility (i.e. costs) would be an important aspect for ecologists to consider.

Now included: lines 323-326.

Point 2. "solar-refuge-dataset-meansensor.csv" was not available.

- Different datasets have been provided in an Excel file. However, the csv file mentioned above (in the R script) was not readily available. It is better to provide the csv file than letting reader to locate the data among spreadsheets in Excel file.

Good Catch. It is now available in the figshare depository:
<https://figshare.com/s/5958b330d9ae094db4c8>

Point 3. Headings and subheadings were missing all together between Line 47 and Line 174. the manuscript.

- It is hard to for readers to navigate around the article, there are two different topics (ecology of snails and light sensor) present together here.

As mentioned above, the sensor details are now restricted to the terminal Methods section. Discrete Introduction and Results/Discussion heading are now used.

Point 4. Line 164-174 - "...are physiologically limited by moisture availability and are mostly found in shaded areas."

- Probably a bit more discussion of literature review required to establish the effect of light exposure to the moisture and temperature for the location/habitat.

- This study proves the association between the light intensity and behaviour of the snails but there was no causation can be proved with the existing data.

- Light intensity might not be the direct factor. Temperature, humidity and light intensity are not mutually exclusive factors (See also line 140-142). Further discussion on this would be important.

All good points. We have inserted new introductory text (lines 50-70) that will help the reader put the negative phototaxis characteristic of land snails into a broader context that includes temperature, humidity, shell albedo effects and the respective ecologies of

the tree snail and the predator. There is now a thorough discussion of the shortcomings of our data together with the recommendation that sensors documenting temperature and humidity in addition to light intensity be used (lines-156-170) for future work as well as an explicit statement that our results are correlational and are merely consistent with the hypothesis rather than proving causation (line 175).

Point 5. Line 170 – 171 - “Long-term conservation planning for *Partula hyalina* should involve mapping, protection and maintenance of suitable forest edge habitats in as many Tahitian valleys as possible.”

- Probably this need to be elaborated more regarding to how the finding (1) can improve current conservation strategies.

Done. New text added – see lines 183-195.

Reviewer #2 (Remarks to the Author):

This paper describes a novel way of testing the hypothesis that an invasive predatory snail fails to drive to extinction a native, threatened non-predatory snail because both have different tolerance ranges for exposure to sunlight. The authors have tested this hypothesis by applying a novel, miniature sensor, small enough to be carried by these snails on their shells.

The results of the study certainly suggest that, indeed, *Partula* escapes predation and extinction because it is able to withstand greater insolation than its predator.

The main concern I have with the paper is that the snails' activity was measured in two different ways: in *Euglandina*, the sensors were placed on the snails themselves, whereas in *Partula*, for permit restriction reasons, they were placed near resting snails. Also, if I understand correctly, the measurements were not taken in a habitat where the two species occur together, but in separate locations. This weakens the conclusions somewhat.

Those two concerns are valid and they and their implications are now addressed directly in the revised text (lines 156-170 and 175).

In addition, I have a number of minor, mostly technical comments:

--in various places in the paper, sizes are given as $A \times B \times C$ mm³. I would prefer to use either dimensions ($A \times B \times C$ mm) or volume (Xmm³). A similar remark applies to surface areas ($A \times B$ mm²).

Done.

--I am surprised the paper does not refer to earlier experiments in *Cepaea nemoralis* by Jones (1982), who used a light-sensitive paint to determine the amount of time individuals spent exposed to sunlight (Reference: Jones, J.S., 1982. Genetic differences in individual behaviour associated with shell polymorphism in the snail *Cepaea nemoralis*. Nature, volume 298, pages749–750.)

Good Catch – now included (ref 19).

--Line 76: replace "and" by "that"

Done (line 80).

--Line 77: "principal" instead of "principle"

Done (line 234 – this section has been moved to “Methods”).

--Lines 70-100 are filled with a lot of technical detail about the sensors that were developed. While I appreciate that the new technology requires explanation, perhaps this is a little overdone for the main body of what otherwise is a biology, not an electronics paper.

This entire sensor development section has been moved to “Methods”.

--I think the main text should also explain (as is done in the Methods section) why sensors were not attached to *Partula* snails, but only to *Euglandina* snails.

Done (lines 88-89; also included in Fig 1b Legend: lines 594-598).

Menno Schilthuizen, August 26th, 2019
(I waive anonymity.)

Reviewer #3 (Remarks to the Author):

Dear authors,
thank you for your submission to *Communications Biology*. The article demonstrates the use of a miniaturized sensor to determine the amount of environmental light intensity, which fits well into the hypothesis of exposure to light be a strategy adopted by preys to seek refuge from predators. Considering the size of the animals, the sensor is a perfect solution as it is small (and I believe light).
The article is well-structured, methods well-described, references are relevant and

figures overall good.

I am supportive of this manuscript to be published, provided two small points could be clarified by the authors:

1. Is environmental light intensity the key for survival? Do snails forage at night?

Rationale: I would appreciate if the authors could elaborate more on this. Light incidence is clearly distinctive, but it could be a response to other parameters already reported in the literature (e.g., temperature and humidity). If so, it could make your initial hypothesis ill-designed when solar refuge is not a unique factor, but the environmental conditions including temperature, humidity, wind speed, etc. Therefore, unless more robust arguments are presented in response to this first point, the hypothesis must be reformulated, title and claims changed accordingly.

We have inserted new introductory text (lines 50-70) that will help the reader put the negative phototaxis characteristic of land snails into a broader context that includes temperature, humidity, shell albedo effects and the respective ecologies of the tree snail and the predator. There is now a thorough discussion of the shortcomings of our data together with the recommendation that multi-day sensors documenting temperature and humidity in addition to light intensity for co-occurring populations be used (lines 156-170) for future work as well as an explicit statement that our results are correlational and are merely consistent with the hypothesis rather than proving causation.

2. Consider limitations of the proposed methods. For example:

Weight of the sensor on small snails, smell of pigments or glue in the predator behaviour, is only a few days enough? Age of the animals used in the experiments? When the predators last ate and how frequently they eat?

The main limitations of our method were that 1) we were unable to study co-occurring predators and tree snails (I think we would have eventually found some if we had a few months to work on the island), 2) we did not have permission to attach sensors to the tree snails using our method, 3) (with one accidental exception) we limited our data collection to a single day per snail and 4) we looked at only one environmental variable (ambient light intensity). These are now addressed in the revised text (lines-156-170).

Regarding the specific questions, we targeted active, adult predators only (now explicit in line 95) and, being wild organisms, we did not know their previous individual feeding histories (lines 164-166). However, *Euglandina rosea* is an active predator that engages in prolonged searches for prey mucus trails to follow (see new Davis-Berg reference # 31) and it is reasonable to assume that the snails we targeted were actively hunting – they certainly looked the part! We prototyped sensor attachment methods in Michigan using the smaller snail *Cepaea nemoralis* (Extended Data Figure 9) and, in these pilot experiments, it was clear that sensor attachment did not significantly impact their behavior or locomotion – they very quickly resumed normal activities. That also seemed to be the case with *E. rosea* on Tahiti: they promptly resumed active crawling

and this is now detailed in the text (lines 96-104, 167-168; Figure 1c, Extended Data Figures 2 & 3).

Overall the manuscript is well-written. I would only suggest removing adjectives such as "potent" (line 142), consistent use of commas to express numbers above 1,000 (e.g., check lines 147 and 148) and the constant use of "due" and "due to" instead of less colloquial terms such as "resulting from" and "because of".

Done.

I hope the authors would find my considerations useful and will address the points I raised and recommendations I provided.

Yes, they are useful - thank you.

Best regards,

Paulo de Souza

REVIEWERS' COMMENTS:

Reviewer #1 (Remarks to the Author):

Authors have responded and improved the manuscript based on my comments from the previous reviewer reports. I find the article to be written in a clear and concise manner. The research methods used in this article are novel and the conclusions are supported by the data. I do not have further comments.

Reviewer #3 (Remarks to the Author):

Dear authors,
thank you again for considering the recommendation provided in my first review. I am pleased to see you addressed all comments.
I made a recommendation to the editor to accept your submission in its current form.
Kind regards,
Paulo de Souza